# Spatio-Temporal Analysis of Carbon Sequestration in Different Ecosystems of Iran and Its Relationship with Agricultural Droughts

**Muhammad Kamangar [1,2], Ozgur Kisi [3,4] and Masoud Minaei [1,2,*]**

1. Department of Geography, Ferdowsi University of Mashhad, Mashhad 917794883, Iran
2. Geographic Information Science/System and Remote Sensing Laboratory (GISSRS: Lab), Ferdowsi University of Mashhad, Mashhad 917794883, Iran
3. Department of Civil Engineering, University of Applied Sciences, 23562 Lübeck, Germany
4. Department of Civil Engineering, Ilia State University, 0162 Tbilisi, Georgia
* Correspondence: m.minaei@um.ac.ir; Tel.: +98-9355471477

**Abstract:** The increase in environmental and human-related changes (e.g., increase in the carbon cycle flux of plants) has increased the dynamism of ecosystems. Examining fluctuations in net primary production (NPP) is very important in adopting correct strategies for ecosystem management. The current study explores the spatiotemporal variations in NPP and its association with agricultural droughts in Iran's ecosystems over 20 years (2000–2020). Mann–Kendall and Sen's slope methods in each pixel were used to track changes in trends. Drought upsets the terrestrial carbon cycle balance. In this study, Vegetation Health Index (VHI) used to assess drought that extracted from different bands of images satellite. Then, the relationship between NPP rates and agricultural droughts was investigated through running Pearson correlation. The results demonstrated that Iran's annual share of carbon sequestration is 1.38 kg*C/m²/year. The highest carbon sequestration rate was recorded in Caspian Hyrcanian forests. In contrast, the lowest rate was observed in the Arabian Desert and East Sahero-Arabian xeric shrublands in southwestern Iran. Moreover, the highest photosynthesis variations were recorded in Arabian Desert and East Sahero-Arabian xeric shrublands and Tigris–Euphrates alluvial salt marsh, while the lowest changes were registered in Badghyz and Karabil. In total, 34.2% of the studied pixels showed a statistically significant rising or falling trend. Sen's slope estimator demonstrated that the sharpest negative trend in carbon sequestration belonged to Caspian Hyrcanian mixed forests ($-12.24$ g*C/m²/year), while the sharpest positive trend was observed in Azerbaijan shrub desert and steppe (12.29 g*C/m²/year). The results of the Pearson correlation revealed significant correlations between NPP and VHI in different ecosystems with coefficients ranging from $-0.93$ to 0.95. The largest area with a positive correlation (33.97%) belonged to the Zagros Mountains forest steppe. Identification of areas with the greatest carbon sequestration changes could result in prioritizing varied ecosystems management for carbon sequestering. It can be also utilized in environmental planning such as scaling up ecosystem values or estimating current and past ecological capacity.

**Keywords:** carbon cycle; Hyrcanian forest; MODIS; Pearson correlation; trend analysis; vegetation health

## 1. Introduction

Climate change and human activities have influenced the dynamics of terrestrial ecosystems worldwide [1], which in turn has led to changes in the cycle of elements. Carbon is a critical element for all life forms on Earth [2], which regulates the climate and is the primary source of fuel for energy in a global economy [3]. Carbon cycle is a biogeochemical cycle describing the continuous process of carbon composition and release in the biosphere, pedosphere, hydrosphere, geosphere, and atmosphere. In this

process, energy and heat are constantly stored and dissipated [4]. There is a fast (short-term) and a slow (long-term) carbon cycle, and the difference between the two lies in the storage type and the length of time required to complete the cycle. The slow carbon cycle encompasses a longer interval of several million years; the fast carbon cycle, on the other hand, entails carbon activity and return in the soil, water, and atmosphere accomplished through photosynthesis, breathing, and decomposition of living organisms. This cycle may last from several minutes to several years [5]. In recent years, the carbon cycle balance has undergone notable changes caused by the growing consumption of fossil fuels and considerable carbon release in the atmosphere.

Climate change and future climate conditions will cause more and more droughts [6]. Drought is an extreme climate phenomenon that may occur in different climate zones. In recent years, this phenomenon occurs more frequently due to the rising trend in temperature [7]. Droughts are likely to happen in various climate periods. They influence the economy by reducing water and food resources [8]. The increasing frequency of droughts profoundly impacts the vegetation growth pattern and distribution of terrestrial ecosystems [9]. Drought can also alter the intensity of photosynthesis by decreasing the water content of chlorophyll and vegetation [10], and exert a significant effect on NPP patterns. Indeed, the amount of carbon sequestrated through photosynthesis comprises the most significant carbon flux between the terrestrial biosphere and the atmosphere [11].

Terrestrial ecosystems constitute a significant source of the global carbon cycle, absorbing carbon from the atmosphere and slowing down the increase in $CO_2$ concentration [12]. NPP, which is defined as the net rate of carbon production in vegetation for a given period, indicates the mutual impact of environmental factors (e.g., soil and type of tree), human activities, and climatic factors (e.g., temperature, rainfall, and relative humidity) [13,14]. NPP is an essential index to measure ecosystem reaction to climate changes [15]. Generally, climate change boosts NPP in ecosystems, while human activities slightly decrease it [1].

Although it is highly critical to measure carbon flux on a global scale with the aim of adopting policies to reduce climate change and carbon production, such measurement is conducted only in a limited number of areas that are not evenly distributed across the world [16]. To date, no comprehensive supervision method or accurate model has been proposed to gauge the NPP rate. The common methods currently employed to measure NPP mainly include the use of chambers [17–19], the eddy covariance [20–22], and remote sensing [21,23,24]. In chamber method, the NPP of a product is measured by assessing $CO_2$, product flux, and soil. This method requires a massive number of chambers, which constrains NPP measurements on a large scale [25]. The eddy covariance can automatically monitor $CO_2$ exchange but cannot distinguish between soil and product $CO_2$ fluxes. In this method, therefore, it is still necessary to gauge the soil $CO_2$ flux to assess changes in product NPP [21]. Lieth and Whittaker (1975) carried out the first NPP assessment on a global scale through conducting a regression analysis based on the data related to temperature and simple annual measurement of actual evapotranspiration in millimeters gleaned from around 1000 weather stations. Placing the data in an experimental equation and conducting interpolation, they estimated global NPP, which was equal to 118 million metric tons per year of biomass [26]. Over the past five decades, remote sensing on spatial and temporal scales has played an important role in quantifying carbon flux and estimating biomass reserves [27–31]. Some studies which have used this method include [32–36], which examined the impact of global changes on the dynamism of carbon flux in plants.

Previous studies have primarily concentrated on the effect of precipitation and temperature on NPP changes in climatic boundaries. In these studies, the study areas have been initially classified in light of climatic features. Then, the influence of changes in climatic elements on NPP has been explored based on geographical locations [35,37]. Some studies have also examined changes in a particular ecosystem [38–40].

There is a considerable spatial and temporal variation in NPP across the globe, which is basically influenced by climate, land cover, and land use practices [28]. It is essential to conduct further studies in other areas to draw a more comprehensive picture and make

better decisions. The first step in this regard is identifying fluctuations in ecosystem NPP patterns, followed by monitoring and extracting spatiotemporal variation patterns.

Ecologically, two major landscapes in Iran are desert and mountain. As one of the largest countries in world with complex topography, distinct climate areas, and massive plant and animal diversity, Iran supports various ecosystems, which often plays an important role in preserving biodiversity throughout the world. For instance, North Forests and coral reefs in south coastal regions have built a lot of ecosystems and great genetic diversity. Additionally, several Iranian wetlands are globally significant, hosting large populations of migratory birds for wintering. Considering the spatial extent, the climatic diversity distribution of NPP is very high in Iran, particularly in the northern and northwestern parts of the country. In contrast, the eastern and southeastern regions have lower NPP values such that the Northern Forests NPP amounts to 1.3, and in the case of central deserts, it is almost 0. During recent years, the major quantitative changes in NPP are caused by the pure effects of climate changes and intensified anthropogenic activities in Iran. Given the spatial extent, climatic diversity, and lack of relevant data in Iran, few studies have focused on NPP in this country. In addition, there is no comprehensive study on spatiotemporal variations in NPP in different ecosystems and the impact of droughts on this index. To partially address this gap, the primary objective of this study was to delineate spatiotemporal variation patterns in terrestrial carbon flux of plants in different ecosystems of Iran using remote sensing and statistical procedures. The study also aimed to examine the impact of carbon flux variations on agricultural droughts. The findings can help authorities make better decisions regarding risk management to reduce fluctuations in carbon flux.

## 2. Materials and Methods

### 2.1. Study Area

The study area, Iran, is located in Southeast Asia with latitudinal coordinates of 25°–40° N and longitudinal coordinates of 44°–64° E. It covers an area of 1,648,195 km$^2$ [41]. Iran borders Armenia and Azerbaijan in the northwest, the Caspian Sea in the north, Turkmenistan in the northeast, Afghanistan and Pakistan in the east, the Persian Gulf and the Oman Sea in the south, and Iraq and Turkey in the west [42] (Figure 1). Iran is the only country in the world with coastlines on the Caspian Sea, the Persian Gulf, and the Indian Ocean [43]. In general, Iran is located in a mountainous and semi-arid region and its average height is over 1200 m above sea level. Since Iran encompasses a vast area, hosts numerous geographical factors, and is located at the transition point of different atmospheric circulation systems, it is home to a wide range of climates and ecosystems. Moreover, due to its biological historical background and considerable power in speciation, the country enjoys much biodiversity. According to the latest census, Iran's population exceeds 85 million with an annual population growth rate of 1.2% [44]. In 2019, Iran recorded a $CO_2$ per capita emission rate of 7.5 tons, which is over 1.6 times the global average. Its emission growth rate since 1990 is more than 134% [45], indicating an excessive rate of carbon emission and poor environmental conditions.

### 2.2. Methods

In order to conduct spatiotemporal analysis of carbon sequestration in Iran's ecosystems and explore its association with agricultural droughts, NPP data and VHI were analyzed for the entire country from the spatiotemporal perspective. Then, trends and breaks in the data were analyzed using Mann–Kendall and Sen's slope methods. The identified trends in each of the ecosystems were assessed from the spatiotemporal perspective. After that, the relationship between carbon sequestration and drought was assessed. Figure 2 illustrates the overall procedure followed in this study. In what follows, further details are provided for those sections requiring more elaboration.

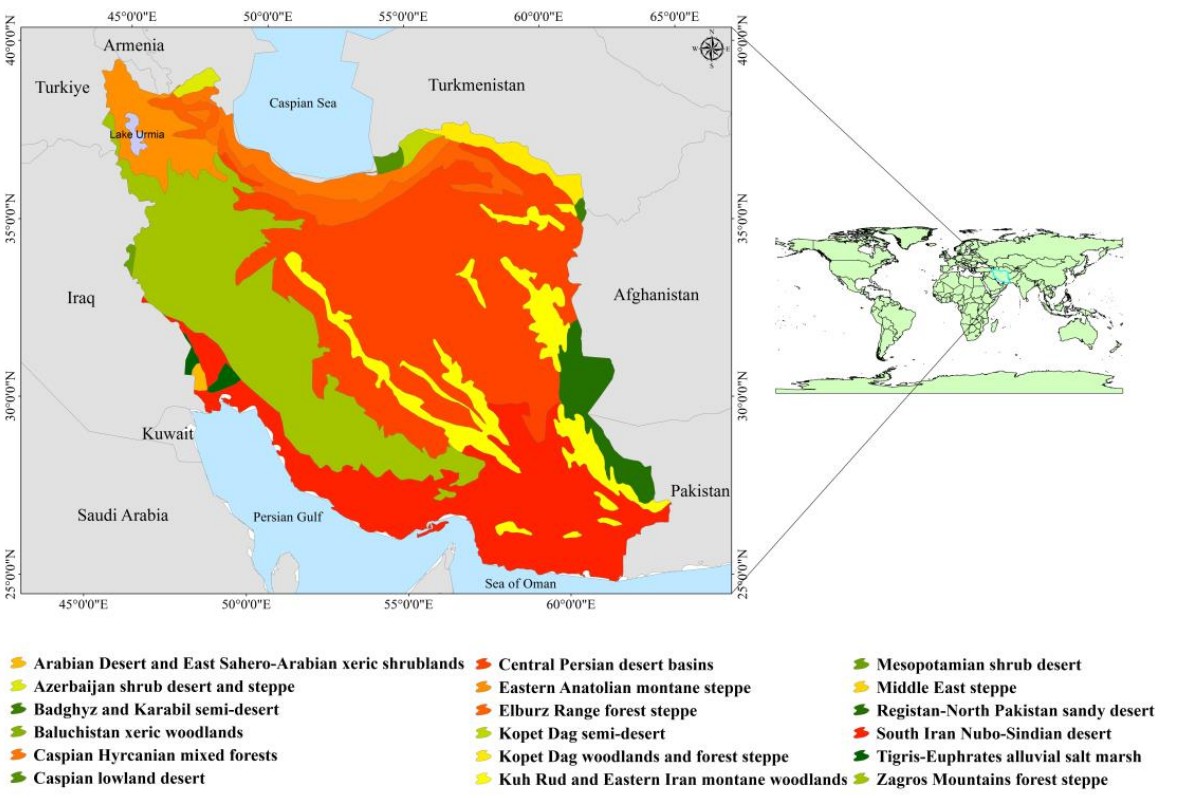

**Figure 1.** Geographical location of Iran and its ecosystems. Ref: developed by USGS 2020 (https://rmgsc.cr.usgs.gov/outgoing/ecosystems/, accessed on 26 October 2022).

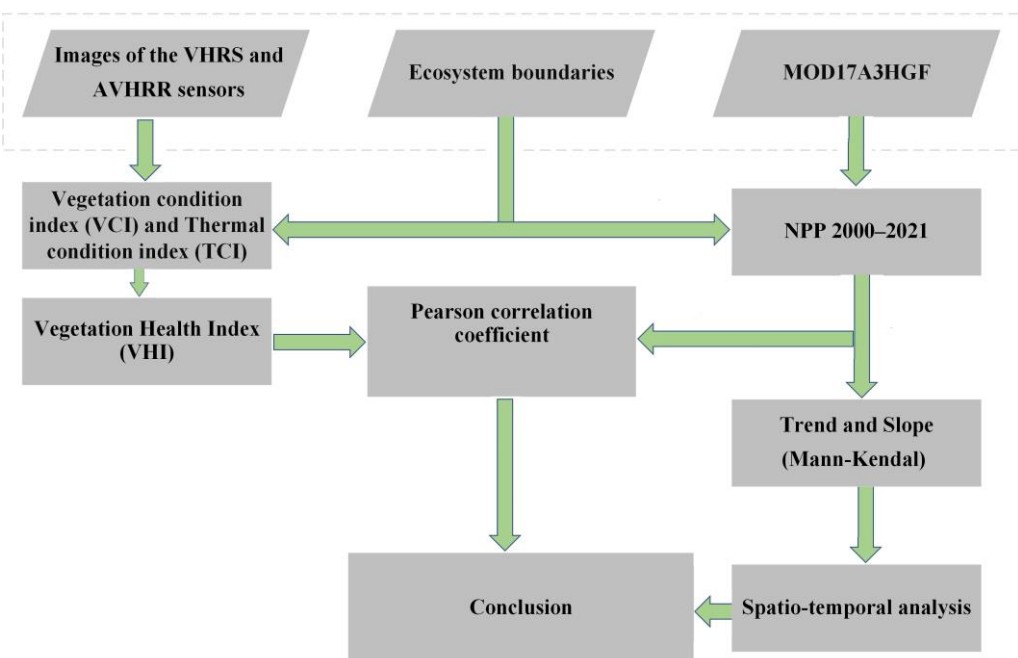

**Figure 2.** Flowchart of methodology.

### 2.2.1. Calculation of Net Primary Production (NPP)

The Monteith equation [46] was used to estimate *NPP* based on satellite images. According to Equation (1), annual *NPP* is obtained from gross primary productivity (*GPP*).

To this end, daily gross *NPP* was first calculated and then the respiration rate required for maintenance and growth of the plant organs was subtracted, yielding NPP [47–49].

$$NPP = \sum_{1}^{365} GPP - R_m - R_g \tag{1}$$

where *GPP* is the gross primary productivity rate, *Rm* is the residual plant respiration, and *Rg* is the respiration rate required for plant growth. $R_m - R_g$ was calculated based on leaf area index (LAI) product, MODIS sensor (MODIS 15), and climate data [48]. MOD17A3HGF V6 product provides information about annual NPP at 500 m pixel resolution from the given year, which was retrieved from https://lpdaac.usgs.gov, accessed on 18 October 2022, maintained by the NASA EOSDIS Land Processes Distributed Active Archive Center (LP DAAC) at the USGS Earth Resources Observation and Science (EROS) Center.

### 2.2.2. Calculation of Agricultural Drought Index

Vegetation Health Index (VHI) combines both the Vegetation Condition Index (VCI) and the Temperature Condition Index (TCI), and is used to assess drought stress [50] based on Equation (2).

$$VHI = \alpha VCI + (1 - \alpha)\, TCI \tag{2}$$

where $\alpha$ is a fixed coefficient equal to 0.5, and VCI and TCI were calculated using the following Equations. In these equations, *NDVI* and *LST* are Normalized Difference Vegetation Index and Land Surface Temperature, respectively, which were extracted from different bands of MODIS images based on Equations (3) and (4).

$$VCI = \frac{NDVI - NDVI_{min}}{NDVI_{max} - NDVI_{min}} * 100 \tag{3}$$

$$TCI = \frac{LST_{max} - LST_i}{LST_{max} - LST_{min}} * 100 \tag{4}$$

VHI values were divided into five categories: very severe drought (10<), severe drought (10–20), moderate drought (20–30), mild drought (30–40), and no drought (>40) [50]. VHI with a resolution of 4 km and a weekly time scale was selected from the Blended Vegetation Health Product (Blended-VHP). It is widely used to supervise and identify agricultural droughts [50–53]. Blended-VHP products were derived from the satellite images of the Visible Infrared Imaging Radiometer Suite (VIIRS) (since 2013) and the Advanced Very High-Resolution Radiometer (AVHRR) (1981–2012). In the current study, arithmetic mean was used to obtain annual VHI.

### 2.2.3. Trend Analysis

Trend analysis can be carried out using both non-parametric and parametric tests [54]. Normal distribution of the data set is not a prerequisite for conducting non-parametric tests. Moreover, such tests are highly insensitive to outliers; thus, they are widely used by researchers. Mann–Kendall is one of the most commonly used non-parametric tests for detecting trends in time series data [55–57].

This test was initially used by Mann in 1945 and subsequently developed by Kendall in 1970 [58,59]. The null hypothesis in this test states that the data are randomly distributed and do not form any trend. Rejecting the null hypothesis confirms the presence of a trend in the data set. First, the differences between each observation and the other ones were calculated and then parameter S was obtained using Equation (5).

$$S = \sum_{k=1}^{n-1} \sum_{j=k+1}^{n} sgn(x_j - x_k) \tag{5}$$

where n is the number of observations, $x_j$ and $x_k$ show the *j*th and *k*th series, respectively. The *sgn* function was calculated as follows:

$$sgn(x_j - x_k) = \begin{cases} 1 \; if \, (x_j - x_k) > 0 \\ 1 \; if \, (x_j - x_k) = 0 \\ 1 \; if \, (x_j - x_k) < 0 \end{cases} \tag{6}$$

The values of *S* and *V*(*S*) were used to compute the test statistic *Z* as follows:

$$Z = \begin{cases} x = \frac{S-1}{\sqrt{var(s)}} \\ x = \frac{S+1}{\sqrt{var(s)}} \end{cases} \qquad \begin{cases} if \quad S > 0 \\ if \quad S = 0 \\ if \quad S < 0 \end{cases} \tag{7}$$

If |*Z*| is larger than *Z* critical value, then the null hypothesis is invalid, indicating that the trend is significant.

All the abovementioned steps were coded in the R programming platform. All NPP images during 2000–2020 were recalled. Then, the data of each pixel were extracted as a matrix, matched with the ecosystem boundaries, and analyzed. The magnitude of a trend can be estimated using Sen's slope estimator, which is a non-parametric method [60,61]. Equation (8) was used to calculate Sen's estimate:

$$f(1) = Qt + C \tag{8}$$

where C is a constant and Q is the trend obtained through the following equation:

$$Q_{ij} = \frac{X_j - X_k}{j - k} \tag{9}$$

where *Q* indicates the trend magnitude, $X_k$ and $X_j$ are the data values in the time series, and $Q_i$ is the Sen's slope estimate obtained from the medians of the A number of observations (N). The number of odd observations is obtained using Equation (10).

$$Qmed = Q\left(\frac{N+1}{2}\right) \tag{10}$$

The number of even observations is obtained through Equation (11).

$$Qmed = \frac{1}{2}\left(Q_{[\frac{N}{2}]} + Q_{[\frac{N+2}{2}]}\right) \tag{11}$$

The confidence interval is also calculated from the specific probability of determining whether the median trend is statistically different from zero. It is calculated using Equation (12).

$$Ca = Z_{1-\frac{a}{2}}\sqrt{Var(s)} \tag{12}$$

where $Z_{(1-a/2)}$ is typically obtained from the standard normal distribution table [62].

2.2.4. Correlation Analysis

Pearson correlation coefficient indicates the correlation of dispersion. It shows the direction and the strength of correlation and is calculated based on Equation (13) [63].

$$r_{xy} = \frac{\sum\limits_{i=1}^{N}(X_i - \overline{X})(Y_i - \overline{Y})}{\sqrt{\sum\limits_{i=1}^{N}(X_i - \overline{X})^2 \sum\limits_{i=1}^{N}(Y_i - \overline{Y})^2}} \tag{13}$$

The equation above yields an output that ranges from −1 to +1, with the former indicating a perfect negative correlation between the two variables and the latter demonstrating a perfect positive correlation. If the correlation between the variables is not perfect, the correlation coefficient falls between −1 and +1. To assess the null hypothesis, the correlation coefficient is compared against the t distribution with n − 2 degrees of freedom in light of the desired significance level. If the absolute value of the observed t is greater than the critical t, the null hypothesis is rejected.

## 3. Results

The annual NPP distribution pattern in different ecosystems for the interval ranging from 2000 to 2020 was calculated. The results showed that the degree of carbon sequestration in Iran, which varies to 1.38 kg*C/m$^2$, is a function of geographical longitude and latitude (Figure 3). In fact, the northern parts of the country registered a higher amount of carbon sequestration in comparison with the southern regions. There is also a decrease in the amount of carbon sequestration in the direction from west to east, with the exception of a cluster in the northern regions, where a high degree of carbon sequestration was detected due to the existence of the Hyrcanian coniferous forest ecosystem on the edge of the Caspian Sea. Almost no carbon sequestration was recorded in the central parts of Iran, which is home to the ecosystem of the Central Persian desert basins and is characterized by lack of rainfall and poor vegetation. Ecosystems which recorded an average carbon sequestration above 100 g*C/m$^2$ are located in the north of Iran, whereas those registering an average carbon sequestration of fewer than 100 g*C/m$^2$ are mainly located in the central and eastern parts of the country. The highest amount of carbon sequestration was detected in Caspian Hyrcanian mixed forests, Azerbaijan shrub desert and steppe, and Elburz Range forest steppe, which is located in the north of Iran, with average carbon sequestration rates of 492.90, 329.51 and 225.99 g*C/m$^2$, respectively. Conversely, Arabian Desert and East Sahero-Arabian xeric shrublands and Tigris–Euphrates alluvial salt marsh, located in southwestern Iran, were the ecosystems that recorded the lowest rate of carbon sequestration (0.20 and 12.60 g*C/m$^2$, respectively).

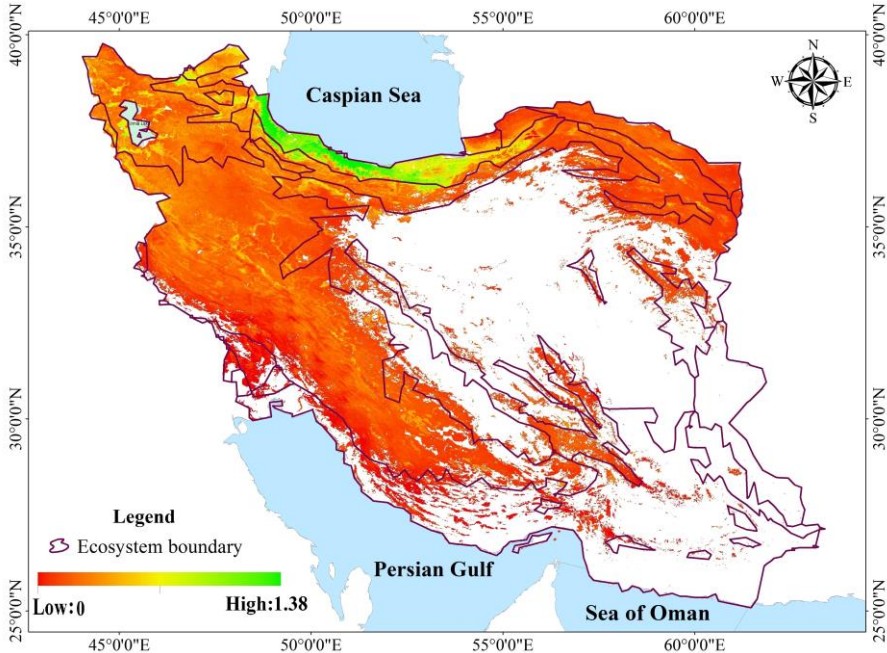

**Figure 3.** The spatial distribution of median net primary productivity in Iran, 2000–2020.

We also calculated the coefficient of variation (CV) for different ecosystems. Arabian Desert and East Sahero-Arabian xeric shrublands (CV = 5.99) and Tigris–Euphrates alluvial salt marsh (CV = 1.87) in southwestern Iran were the two ecosystems registering the

highest variability, which indicates instability in the amount of photosynthesis in these ecosystems. On the other hand, the lowest CVs were recorded for the Middle East steppe and the Badghyz and Karabil semi-desert ecosystem (with CV indices of 0.19 and 0.20, respectively). This indicates that vegetation photosynthesis in the studied period was relatively stable in these regions during the studied period. Moran's I was calculated in light of the values of each pixel to understand whether carbon sequestration followed a clustered or dispersed pattern. Given that the value obtained for spatial Moran (0.9701) was higher than the expected value (−0/00001) and the *p*-value was equal to 0.000, the spatially clustered pattern of carbon sequestration in Iran was confirmed. Figure 4 demonstrates the available trends of carbon sequestration for each pixel obtained through conducting Mann–Kendall with a confidence level of 95%. Overall, 65.8% of the pixels in Iran indicates a statistically insignificant trend, while 34.2% of them demonstrates a significant increasing and decreasing trends. As Figure 4 suggests, some parts of western, northwestern, northeastern, and central Iran possess a significant rising trend. The highest increasing trend obtained through the Z score (0.94) was observed in the Eastern Anatolian Mountain steppe. Additionally, the increasing trend of carbon sequestration was detected around mega cities such as Tehran, Isfahan, and Mashhad, which is attributed to planting trees in the surrounding areas of these cities. The highest increasing rates were observed in the Eastern Anatolian Mountain steppe (91.1%) and mountainous Zagros forests (62.26%). Figure 4 further shows that northern and southwestern parts of Iran recorded a significant decreasing trend. The highest declining trend with a Z score of −0.86 was observed in the Caspian Hyrcanian mixed forests ecosystem.

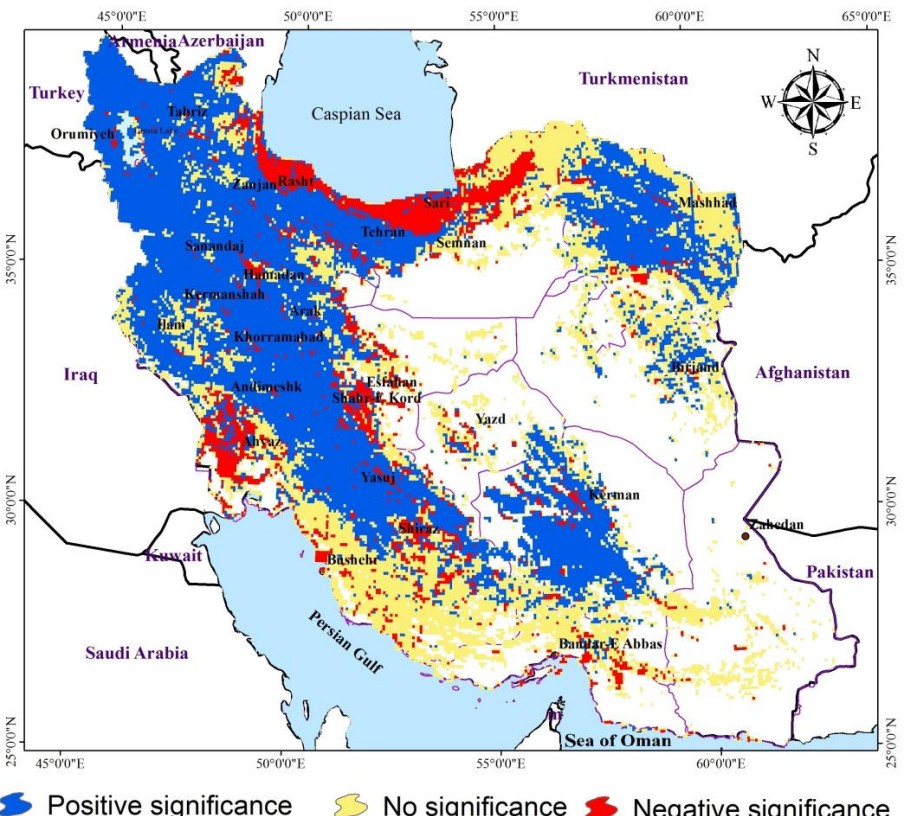

**Figure 4.** Spatial pattern of NPP trends.

Sen's slope method indicated that the spatial distribution of NPP trend changes ranged from 53.41 to −80.73 g*C/m$^2$ (Figure 5). The lowest mean of negative trend (−12.24 g*C/m$^2$) was recorded in the north of Iran in the Caspian Hyrcanian mixed forests ecosystem (Table 1).

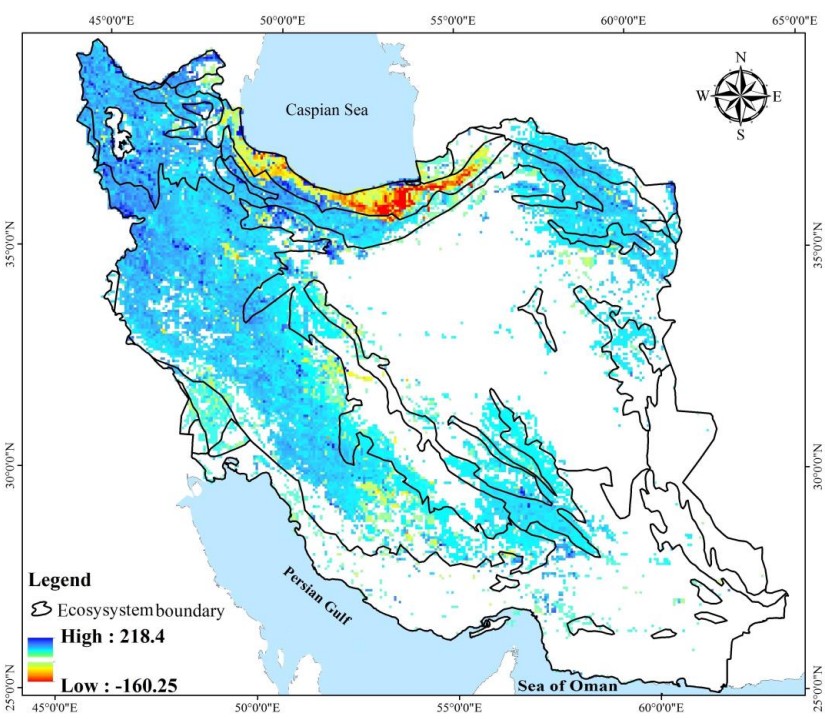

**Figure 5.** Sen's slope of NPP.

**Table 1.** Statistical calculations NPP of slope change in different ecosystems.

| ECO_NAME | MIN (g*C/m$^2$) | MAX (g*C/m$^2$) | RANGE (g*C/m$^2$) | MEAN (g*C/m$^2$) | STD [1] |
|---|---|---|---|---|---|
| Caspian Hyrcanian mixed forests | −80.73 | 52.19 | 132.9 | −12.25 | 16.43 |
| Tigris–Euphrates alluvial salt marsh | −16.39 | 11.58 | 27.98 | −3.09 | 4.73 |
| Arabian Desert and East Sahero-Arabian xeric shrublands | −8.66 | 0.56 | 9.22 | −0.62 | 1.48 |
| Kopet Dag semi-desert | −15.93 | 12.14 | 28.08 | −0.61 | 8.59 |
| Caspian lowland desert | −23.81 | 25.39 | 49.21 | −0.02 | 10.97 |
| Registan–North Pakistan sandy desert | −6.21 | 2.34 | 8.55 | 0.90 | 2.04 |
| South Iran Nubo-Sindian desert and semi-desert | −22.70 | 37.69 | 60.40 | 1.88 | 5.6 |
| Kuh Rud and Eastern Iran montane woodlands | −21.60 | 33.95 | 55.56 | 3.382 | 2.37 |
| Badghyz and Karabil semi-desert | −4.30 | 9.92 | 14.22 | 3.99 | 1.34 |
| Mesopotamian shrub desert | −3.04 | 10.82 | 13.86 | 4.39 | 2.30 |
| Central Persian desert basins | −34.84 | 44.67 | 79.51 | 4.73 | 4.41 |
| Lake | −9.90 | 18.45 | 28.35 | 5.41 | 4.20 |
| Elburz Range forest steppe | −38.08 | 53.15 | 91.24 | 5.59 | 6.28 |
| Zagros Mountains forest steppe | −59.83 | 53.41 | 113.25 | 6.10 | 4.19 |
| Kopet Dag woodlands and forest steppe | −13.26 | 24.57 | 37.84 | 6.19 | 2.69 |
| Middle East steppe | 7.72 | 7.72 | 0 | 7.72 | 0 |
| Eastern Anatolian montane steppe | −78.40 | 51.93 | 130.33 | 8.58 | 4.86 |
| Azerbaijan shrub desert and steppe | −44.71 | 40.42 | 85.14 | 12.29 | 5.79 |

[1] Standard Deviation.

The minimum trend change (−82 g*C/m$^2$) was also observed in this ecosystem. On the contrary, the highest mean of positive trend (12.29 g*C/m$^2$) was detected in the Azerbaijan shrub desert and steppe, while the maximum trend change (53.41 g*C/m$^2$) was registered for the Zagros Mountains forest steppe.

Table 2 displays the percentage and area of trend changes for each ecosystem (kg/m$^2$/year) categorized into four classes: extreme negative trend (−83 to −30), slight negative trend (−30 to 0), extreme positive trend (0 to 30), and slight positive trend (30 to 53).

Caspian Hyrcanian mixed forests had the largest area with negative trend covering (12,598.61774 km$^2$ equal to 24%), whereas the smallest area with negative trend was

recorded for the Badghyz and Karabil semi-desert ecosystem with an area smaller than 1 km$^2$ (0.037%). On the other hand, the Zagros Mountains forest steppe with an area of 149421/1033 (42%) and Kuh Rud and Eastern Iran mountain woodlands with an area covering less than 1 km$^2$ (0.0005%), respectively, were the largest and smallest regions with positive trend. Considering the association between carbon sequestration and agricultural droughts, the regions where a significant positive correlation was observed between NPP and VHI are illustrated in Figure 6. The strongest correlations were detected in the north, northeast, and southwest, while the weakest correlations were recorded in the central parts of Iran. As observed, save for two ecosystems (i.e., Middle East steppe and Baluchistan xeric woodlands), positive correlations between NPP and VHI were observed for the entire country. The strongest and weakest correlations, respectively, belonged to the Zagros Mountains forest steppe (r = 97.33) and the Registan–North Pakistan sandy desert (r = 0.0059).

**Table 2.** Percentage of trend changes for each ecosystem.

| ECO_NAME | −83 to −30 | −30 to 0 | 0 to 30 | 30 to 53 |
|---|---|---|---|---|
| Caspian Hyrcanian mixed forests | 6.42 | 24.64 | 11.41 | 0.13 |
| Tigris–Euphrates alluvial salt marsh | Non-significant | 4.519 | 0.73 | Non-significant |
| Arabian Desert and East Sahero-Arabian xeric shrublands | Non-significant | 3.27 | 0.04 | Non-significant |
| Kopet Dag semi-desert | Non-significant | 0.32 | 0.316 | Non-significant |
| Caspian lowland desert | Non-significant | 2.32 | 1.55 | Non-significant |
| Registan–North Pakistan sandy desert | Non-significant | 0.00 | 0.03 | Non-significant |
| South Iran Nubo-Sindian desert and semi-desert | Non-significant | 0.58 | 1.34 | 0.00 |
| Kuh Rud and Eastern Iran montane woodlands | Non-significant | 0.33 | 18.12 | 0.0 |
| Badghyz and Karabil semi-desert | Non-significant | 0.03 | 20.17 | Non-significant |
| Mesopotamian shrub desert | Non-significant | 0.07 | 6.40 | Non-significant |
| Central Persian desert basins | 0.00 | 0.29 | 8.50 | 0.00 |
| Lake | Non-significant | 0.06 | 1.187 | Non-significant |
| Elburz Range forest steppe | 0.04 | 1.94 | 37.77 | 0.03 |
| Zagros Mountains forest steppe | 0.003 | 1.01 | 42.57 | 0.01 |
| Kopet Dag woodlands and forest steppe | Non-significant | 0.02 | 23.27 | Non-significant |
| Middle East steppe | Non-significant | Non-significant | 4.43 | Non-significant |
| Eastern Anatolian montane steppe | 0.02 | 0.34 | 63.40 | 0.03 |
| Azerbaijan shrub desert and steppe | 0.02 | 0.32 | 32.72 | 0.31 |

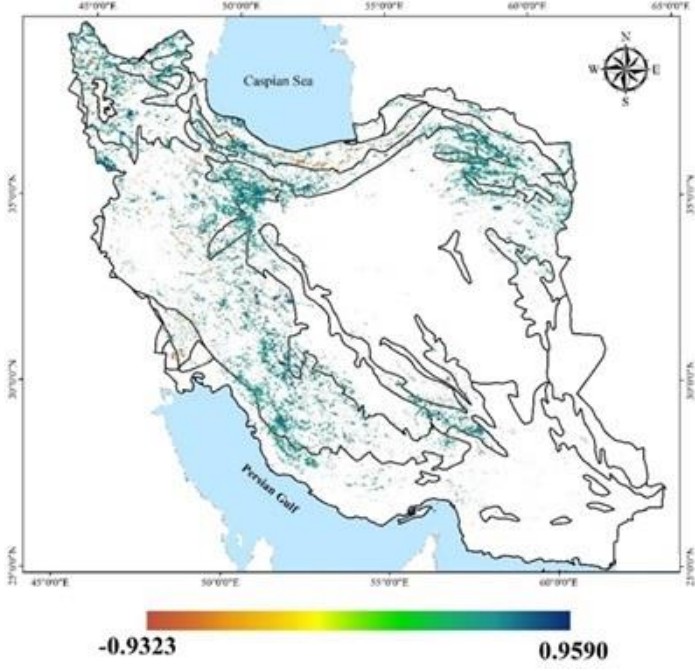

**Figure 6.** Significant correlations of NPP with agricultural droughts.

## 4. Discussion

There is limited (if any) access to ground data on NPP measurement in Iran. Therefore, it is difficult to conduct spatial matching and validate NPP estimates obtained through remote sensing. Previous researches (e.g., [49,59,64,65]) indicate that MODIS data have acceptable accuracy. MODIS sensor-based NPP data are available for only 46% of the areas of Iran's ecosystems. There are no such data for Central Persian desert basins and South Iran Nubo-Sindian desert and semi-desert, which are characterized by a lack of vegetation. In these locations, the degree of carbon sequestration can be accessed via experimental models or ground observations. The average share of terrestrial carbon sequestration of plants in Iran was estimated to be 1.38 kg*C/m$^2$/year. This estimate is larger than the one proposed by the authors of [35], who studied a time period ending in 2014 and used average pixels.

Unlike some previous research, the median of estimated image values was used in the current study. The numerical values of the pixels in the images are affected by cloud conditions, various land cover conditions, or radiometric errors. Therefore, a mean score may yield a value that is far from the actual focal point. Conversely, the median is less susceptible to the lack of data in a particular time or climate extremes.

Liu et al. [40], Song et al. [66] claimed that forests play a significant role in carbon sequestration through photosynthesis. In a similar vein, the findings of this research showed that the forest ecosystems in the north and northwest of Iran, with broad-leaved evergreen trees, account for a large amount of carbon sequestration. In contrast, the ecosystems containing deserts and salt marshes in central and southern Iran recorded the lowest portion in carbon sequestration due to lack of vegetation. The results of trend analysis revealed no statistically significant trend in more than two-thirds of the available pixels studied in this research. Eastern Anatolian Mountain steppe and the Zagros Mountains forest steppe registered the highest positive trend despite the fact that these ecosystems are undergoing declining rainfall and increasing temperature. The upward trend in these ecosystems, therefore, cannot be connected to climate factors; rather, as pointed out by Wu et al. [67] and Chen et al. [68], the growing trend in carbon sequestration may be attributed to human activities and expanding agriculture area, which play a substantial role in increased vegetation. Our findings in this regard are in conflict with those of Bejagam and Sharma's [1]. The impact of human activities on the NPP rate depends on the location where the activities occur; thus, destruction of vegetation in a particular place may result in reduced NPP, while it may lead to higher NPP in another spot. The range of NPP variations in Iran is estimated to be about 135 g/m$^2$/year, representing severe changes and photosynthesis instability. The highest standard deviation for change in carbon sequestration was observed in the Hyrcanian coniferous forest ecosystem, which cannot be explained solely in light of climatic variables. Indeed, destructive human activities in this ecosystem (e.g., deforestation) have led to substantial decline in carbon sequestration. On the other hand, climate change has resulted in rising photosynthesis and NPP rate. Increasing trends can be attributed to climate change and human activities. In contrast, radical decline trends in the short term can only be justified in light of deforestation caused by human activities. At the global scale, greenhouse gas emissions (CH$_4$ and CO$_2$ in particular) are largely responsible for the Earth's warming [69,70]. Terrestrial habitats are generally a sink for atmospheric carbon sequestration and greenhouse gas flux modulation. Grossi et al. [71] proposed a comprehensive methodological approach to both quantitatively and qualitatively estimating emission and sequestration of greenhouse gases by plants and highlighted the potential of green spaces to mitigate climate change impacts. The results of research revealed that highest weakened carbon sequestration occurs in the Zagros Mountains Forest steppe ecosystem, indicating remarkable changes in the ecosystem NPP values, and carbon sequestration and storage decreased owing to the loss of forest cover. Liu et al. [72] in China and Sahu et al. [73] in India investigated the NPP changes in different biomes, too, and suggested that forest and farming biomes accounted for the greatest NPP fluctuations. In Iran, restoring forest ecosystems and preventing further damage or loss to

them through compatible cropping patterns, avoiding deforestation, and fire management planning could help rise carbon capture and sequestration and to some extent modulate increasing climate change trend. Environmental degradation and land use alterations have been driven by Iran's economic growth which in turn considerably influenced ecological processes, including carbon sequestration. However, the relationship between NPP and the level of anthropogenic activities in Iran is very difficult to understand because most of Iran's production activities are concentrated in the central provinces such as Tehran and Isfahan, while these provinces have high reliance of other regions on imports of raw materials and do not exploit local natural resources.

Drought is a serious environmental threat across Iran, mainly due to the climate change; however, its effects are exacerbated by rapid population growth, inappropriate population distributions, land overharvesting, and poor water management [74]. The results also showed that, at the 95% confidence level, only in 27.27% of the pixels the correlation between NPP and VHI is statistically significant. Figure 7 shows the average percentage of correlation between NPP and agricultural drought in different ecosystems. Various values indicate that the effect of droughts on NPP depends on the ecosystem type. These correlations could be associated with the differences in drought intensity, drought duration and vegetation type. Pei et al. [75] and Yu et al. [76] discovered that the positive correlation between NPP and SPEI was more robust in the ecosystems located in arid and semi-arid climatic regions. Similarly, in the current study, the highest correlation was observed in the Zagros Mountains forest steppe (Figure 7), an ecosystem located in a semi-arid climate with the highest amount of rainfall in winter and spring. On the contrary, the weakest correlation was spotted in the Registan–North Pakistan sandy desert with cold desert climate. This ecosystem encompasses a small area in Iran; hence, caution must be exercised in making claims based on the findings. Yu et al. [76] and Qi et al. [77] exploited SPEI to examine the association between NPP and droughts. A problem with using SPEI is that this index may overestimate the impact of droughts in arid and semi-arid regions [13]. This is especially undesirable in Iran, with over two thirds of its area being located in arid and semi-arid climates. Liu et al. [78] also assert that soil moisture is more important than drought stress in ecosystem production. This important variable, however, is not taken into account by SPEI.

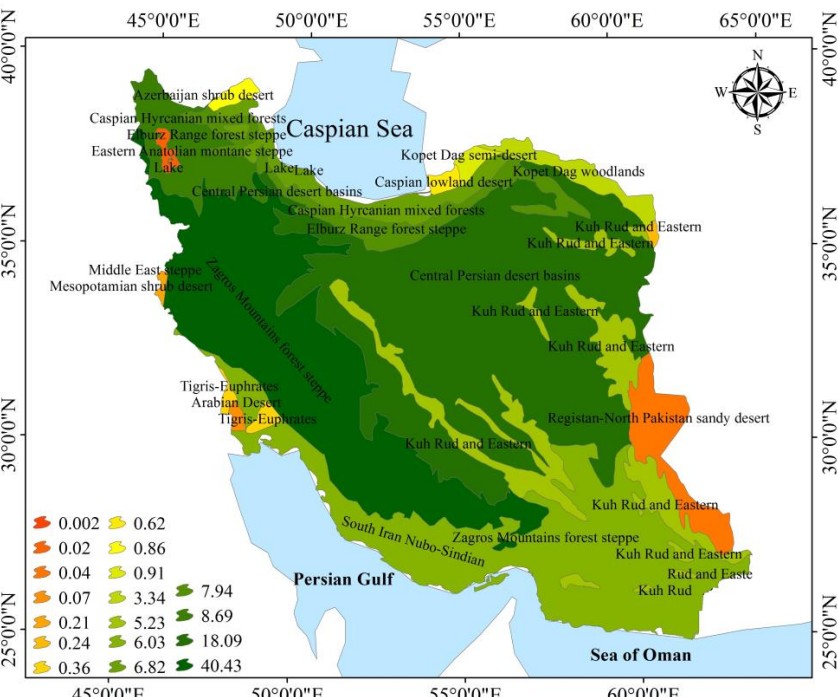

**Figure 7.** Correlation percentage between NPP with agricultural droughts in different ecosystems.

## 5. Conclusions

In this study, satellite images were used to examine spatiotemporal variations in Iran's NPP in light of ecosystem boundaries. Findings showed that the share of carbon sequestration in Iran constitutes 1.38 kg*C/m$^2$/year and follows a clustered spatial pattern. Larger NPP rates were recorded for ecosystems located in the north and northwest of Iran. Furthermore, spatiotemporal variations were not aligned with the ecosystem boundaries; in fact, a decreasing rate of spatial variations was observed in the southward direction. The highest negative trend was detected in the north of Iran, especially in the Caspian Hyrcanian mixed forest ecosystem. Substantially negative carbon sequestration trends were observed in ecosystems with rich vegetation, which supports the idea that measures taken to curb deforestation and natural plant destruction are ineffective. The highest positive trend was spotted in the Azerbaijan shrub desert and steppe ecosystem. This indicates that ecosystems frequently used by humans experience positive trends in carbon sequestration due to agricultural activities. The findings also suggest that NPP rate and agricultural drought have a strong correlation in the Zagros Mountains forest steppe and a weak association in the Registan–North Pakistan sandy desert. However, a significant positive correlation between NPP and agricultural drought was observed. Various correlation coefficients obtained in this study indicate that, besides rainfall and temperature fluctuations, other factors within the ecosystem influence NPP rates. Given the extreme dynamics of climatic factors, it is suggested that future researchers use nonlinear methods (e.g., cross-spectrum analysis of correlations) to identify the degree to which climatic and non-climatic factors influence NPP variations. The findings of this study showed that using remote sensing to monitor carbon sequestration in vast areas, which are influenced by numerous factors, saves a lot of money and time. As such, remote sensing can be used in environmental planning to implement constructive measures. Providing precise information for decision-makers planning in order to balance between society's needs and the exploitation of ecosystems' natural resources during drought periods could contribute to help PNN decreasing. Local non-governmental organizations (NGOs) and their participation in decision-making process, as well as information sharing via social media during drought periods play vital roles in this respect.

**Author Contributions:** Conceptualization, M.K., O.K. and M.M.; methodology, M.K. and M.M.; software, M.K. and M.M.; formal analysis, M.K., O.K. and M.M.; investigation, M.K. and O.K.; resources, M.K.; data curation, M.K.; writing—review and editing, M.K., O.K. and M.M.; visualization, M.K.; project administration, M.M. All authors have read and agreed to the published version of the manuscript.

**Funding:** Ferdowsi University of Mashhad, Grant Number: FUM25143.

**Data Availability Statement:** https://github.com/poyan2021/Carbon-sequestration-.git, accessed on 4 February 2023.

**Acknowledgments:** The authors would like to acknowledge the (LP DAAC) science team to freely provide the data sets used in this study. We acknowledge the Ferdowsi University of Mashhad for support and also thank the Academic Editor for his comments and the anonymous reviewers for their insight.

**Conflicts of Interest:** All authors certify that they have no affiliations with or involvement in any organization or entity with any financial interest or non-financial interest in the subject matter or materials discussed in this manuscript.

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
