# Peer review of "Spatio-Temporal Analysis of Carbon Sequestration in Different Ecosystems of Iran and Its Relationship with Agricultural Droughts"

_sustainability, doi:10.3390/su15086577_

Round 1

Reviewer 1 Report

Plant productivity also plays a major role in the global carbon cycle by absorbing some of the carbon dioxide released when people burn coal, oil, and other fossil fuels. The carbon plants absorb becomes part of leaves, roots, stalks or tree trunks, and ultimately, the soil. Net primary production (NPP) is strictly defined as the difference between the energy fixed by autotrophs and their respiration, and it is most commonly equated to increments in biomass per unit of land surface and time. As such, this study explores the spatiotemporal variations in NPP and its association with agricultural droughts in ecosystems over 20 years (2000-2020) in Iran. The results showed that Iran’s share of carbon sequestration varies greatly and the highest photosynthesis variations are recorded in Arabian Desert and East Sahero-Arabian and Tigris-Euphrates alluvial salt marsh.

Generally, some revision suggestions are listed below:

1) 'Iran’s share of carbon sequestration ranges from 0 to 1.38 kg*C/m2/yr'. There's no point in mentioning zero in NPP.

2) Introduce the importance of Iran in terrestrial ecosystems in Introduction.

3) Show in details the verification in your research otherwise the results of your research is in doubt. 

4) Discuss your research contribution to greenhouse gas emissions reduction from the perspective of climate change.

Author Response

Hello and respect

The authors would like to thank the area editor and the reviewer for their precious time and invaluable comments. We have carefully addressed all the comments. The corresponding changes and refinements made in the revised paper are summarized in our response below.

1) There's no point in mentioning zero in NPP. 'Iran’s share of carbon sequestration ranges from 0 to 1.38 kg*C/m2/yr'.

The results demonstrated that Iran’s annual share of carbon sequestration is 1.38 kg*C/m2/yr.

The results showed that the degree of carbon sequestration in Iran, which varies to 1.38 kg*C/m2, is a function of geographical longitude and latitude.

2) Introduce the importance of Iran in terrestrial ecosystems in Introduction.

Ecologically, two major landscapes in Iran are desert and mountain. Over 8,200 plant species and 1,850 species of animals are recognized in this country. Being as one of the largest countries in world with complex topography, distinct climate areas, and massive plant and animal diversity, Iran supports various ecosystems which often plays an important role in preserving biodiversity throughout the world. For instance, North Forests and coral reefs in south coastal regions have built a lot of ecosystems and great genetic diversity. Additionally, several Iranian wetlands are globally significant; hosting large populations of migratory birds for wintering.

3) Show in details the verification in your research otherwise the results of your research is in doubt. 

The original strategy for validation of the MOD 17 GPP and NPP data can be found in Running et al. 1999. This paper laid the foundation for the interaction between MODLAND and the eddy covariance flux tower community, illustrating the mutual advantage of flux towers providing direct measurement of carbon and water fluxes, while MODIS land products provided a spatial extrapolation of the tower data. It also laid out the conceptual basis for swapping out various components of the MOD17 algorithm in order to isolate points of uncertainty and inaccuracy.

And this estimate is lesser than the one reported by Kamali and others [34], who studied an era ending in 2014 and utilized average pixels. Unlike some previous research, the median of estimated image values was used in the current study. The numerical values of the pixels in the images are affected by cloud conditions, various land cover conditions, or radiometric errors. Therefore, a mean score may yield a value that is far from the actual focal point. Conversely, the median is less susceptible to the lack of data in a particular time or climate extremes. Song and others [65], Liu et al. [39], and Zhang et al. [66] claimed that forests play a significant role in carbon sequestration through photosynthesis. In a similar vein, the findings of this research showed that the forest ecosystems in the north and northwest of Iran, with broad-leaved evergreen trees, account for a large amount of carbon sequestration.

4) Discuss your research contribution to greenhouse gas emissions reduction from the perspective of climate change.

At the global scale, greenhouse gas emissions (CH4 and CO2 in particular) are largely responsible for Earth's warming (Hui et al., 2021; Kumar et al., 2021). Terrestrial habitats are generally a sink for atmospheric carbon sequestration and greenhouse gases fluxes modulation. Grossi et al. (2021) proposed a comprehensive methodological approach to both quantitatively and qualitatively estimating of emission and sequestration of greenhouse gases by plants and highlighted the potential of green spaces to mitigate climate change impacts. The results of research revealed that highest weakened carbon sequestration is in the Zagros Mountains Forest steppe ecosystem indicating remarkable changes in the ecosystem NPP values, and carbon sequestration and storage decreased owing to the loss of forest cover. Liu et al. in China and Sahu et al. in India investigated the NPP changes in different biomes, too, and suggested that forest and farming biomes accounted for the greatest NPP fluctuations. In Iran, restoring forest ecosystems and preventing further damage or loss to them through compatible cropping patterns, avoiding deforestation, and fire management planning could help rise carbon capture and sequestration and to some extent, modulate increasing climate change trend.

5- Is the article adequately referenced? Must be improved

Kumar, A.; Bhatia, A.; Sehgal, V. K.; Tomer, R.; Jain, N.; Pathak, H. Net Ecosystem Exchange of Carbon Dioxide in Rice-Spring Wheat System of Northwestern Indo-Gangetic Plains. Land 2021, 10 (7), 701. doi:10.3390/land10070701

Liu, S.; Ji, C.; Wang, C.; Chen, J.; Jin, Y.; Zou, Z.; Li, SH.; Niu, S.; Jianwen Zou, J. Climatic Role of Terrestrial Ecosystem under Elevated CO2: A Bottom-Up Greenhouse Gases Budget. Ecology Letters 2018, 21 (7), 1108–1118. doi:10.1111/ele.13078

Sahoo, U.K.; Tripathi, O.P.; Nath, A.J.; Deb, S, Das, D.J.; Gupta, A.; Devi, N.B.; Charturvedi, S.S.; Singh, S.L.; Kumar, A., Tiwari, B.K. Quantifying Tree Diversity, Carbon Stocks, and Sequestration Potential for Diverse Land Uses in Northeast India. Frontiers in Environmental Science 2021, 724950. doi: 10.3389/fenvs.2021.724950

Kogan, F.N.; 1995: Application of vegetation index and brightness temperature for drought detection. Advances in Space Research 1995, 15(11): 91–100. DOI: 10.1016/0273-1177(95)00079-T.

Madani, K. Water management in Iran: what is causing the looming crisis .  Journal of Environmental Studies and Sciences volume 2014, 4, 315–328 (2014). https://doi.org/10.1007/s13412-014-0182-z

Hui, D.; Deng, Q.; Tian, H.; and Luo, Y.; Global Climate Change and Greenhouse Gases Emissions in Terrestrial Ecosystems, 1st ed.; Editors Lackner, M.; Sajjadi, B.; Chen, W.Y; New York, NY: Springer: New York, USA, 2021, pp. 216-219, ISBN 978-1-4614-6431-0.https://doi:10.1007/978-1-4614-6431-0_13-3

Nisbet, E.G.; Encyclopedia of Atmospheric Sciences, 2nd ed.; North, G., John Pyle, J., Zhang, F., Eds.; Elsevier Science: Amsterdam, Netherlands, 2015, Volume 1, pp. 196-201, ISBN 9780122270901. https://doi.org/10.1016/B0-12-227090-8/00014-2

Reviewer 2 Report

before analyzing the data authors should take help from statistician. language is not clear. Results and discussion are not in order.

Author Response

Hello and respect
The authors would like to thank the area editor and the reviewers for their precious time and invaluable comments. We have carefully addressed all the comments. The corresponding changes and refinements made in the revised paper are summarized in our response below.

1- Rewrite this abstract. Way of presentation is not clear?

  The increase in environmental and human-related changes (e.g., increase in the carbon cycle flux of plants) has raised the dynamism of ecosystems. Examining fluctuations in net primary production (NPP) is very important in adopting correct strategies for ecosystem management. The current study explores the spatiotemporal variations in NPP and its association with agricultural droughts in Iran’s ecosystems over 20 years (2000-2020). Mann-Kendall and Sen’s slope methods in each pixel were used to track changes in trends. Then, the relationship between NPP rates and agricultural droughts was investigated through running Pearson correlation. The results demonstrated that Iran’s annual share of carbon sequestration is 1.38 kg*C/m2/yr. The highest carbon sequestration rate was recorded in Caspian Hyrcanian forests. In contrast, the lowest rate was observed in the Arabian Desert and East Sahero-Arabian in southwestern Iran. Moreover, the highest photosynthesis variations were recorded in Arabian Desert and East Sahero-Arabian and Tigris-Euphrates alluvial salt marsh, while the lowest changes were registered in Badghyz and Karabil. In total, 34.2% of the studied pixels showed a statistically significant rising or falling trend. Sen’s slope estimator demonstrated that the sharpest negative trend in carbon sequestration belonged to Caspian Hyrcanian mixed forests (-12.24 g*C/m2/year), while the sharpest positive trend was observed in Azerbaijan shrub desert and steppe (12.29 g*C/m2/year). The results of the Pearson correlation revealed significant correlations between NPP and Vegetation Health Index (VHI) in different ecosystems with coefficients ranging from -0.93 to 0.95. The largest area with a positive correlation (33.97%) belonged to the Zagros Mountains forest steppe. Identification of areas with the greatest carbon sequestration changes could result in prioritizing varied ecosystems management for carbon sequestering. It can be also utilized in environmental planning such as scaling up ecosystems' values or estimating current and past ecological capacity.

2- Reference?

Carbon cycle is a biogeochemical cycle describing the continuous process of carbon composition and release in the biosphere, Pedosphere, hydrosphere, geosphere, and atmosphere.

Nisbet, E.G.; Encyclopedia of Atmospheric Sciences, 2nd ed.; North, G., John Pyle, J., Zhang, F., Eds.; Elsevier Science, Amsterdam, Netherlands, 2015, Volume 1, pp. 196-201, ISBN 9780122270901. https://doi.org/10.1016/B0-12-227090-8/00014-2

3- Not in good English. Very hard to understand?

Drought is an extreme climate phenomenon that may occur in different climate regimes because of the increasing trend in temperature [6]. Droughts are likely to happen in various climate periods and may influence the economy through a shortage of water and food resources [7].

Drought is an extreme climate phenomenon that may occur in different climate zones. In recent years, this phenomenon occurs more frequently due to the rising trend in temperature [6]. Droughts are likely to happen in various climate periods. They influence the economy by reducing water and food resources [7].

4- Not clear. Re-write

It is believed that NPP is an important index in assessing ecosystems’ reaction to climate change [14].

NPP is an essential index to measure ecosystems’ reaction to climate changes [14].

5- Is not clear?  Some studies have also examined changes in a particular ecosystem 

Over the past five decades, remote sensing on spatial and temporal scales has played an important role in quantifying carbon flux and estimating biomass reserves. Some studies which have used this method include Running et al. (2004), Piato et al. (2009), Saatchi et al. (2011), Xiao et al. (2014), and Robinson et al. (2018). Moreover, Nemani et al. (2003), Xiao (2014), Smith et al. (2016), Li et al. (2018), Kamali et al. (2020), and Zhou et al. (2022) examined the impact of global changes on the dynamism of carbon flux in plants.

6- Not correct explanation:

There is no comprehensive study…

Ecologically, two major landscapes in Iran are desert and mountain. Being as one of the largest countries in world with complex topography, distinct climate areas, and massive plant and animal diversity, Iran supports various ecosystems which often plays an important role in preserving biodiversity throughout the world. For instance, North Forests and coral reefs in south coastal regions have built a lot of ecosystems and great genetic diversity. Additionally, several Iranian wetlands are globally significant; hosting large populations of migratory birds for wintering. Considering spatial extent, the climatic diversity distribution of NPP is very high in Iran particularly in North and Northwest parts of the country. In contrast, East and Southeast regions have lower NPP values such that Northern Forests NPP amounts to 1.3 and in the case of central deserts it is almost 0. During recent years, the major quantitively changes in NPP is caused by the pure effects of climate changes and intensified anthropogenic activities in Iran. Given the spatial extent, climatic diversity, and lack of relevant data in Iran, few studies have focused on NPP in this country. In addition there is no comprehensive study on spatiotemporal variations in NPP in different ecosystems and the impact of droughts on this index. To partially ad-dress this gap, the primary objective of this study was to delineate spatiotemporal variation patterns in terrestrial carbon flux of plants in different ecosystems of Iran using re-mote sensing and statistical procedures. The study also aimed to examine the impact of carbon flux variations on agricultural droughts. The findings can help authorities make better decisions regarding risk management to reduce fluctuations in carbon flux.

7- Reference??  Vegetation Health Index (VHI)

Kogan, F.N. Application of vegetation index and brightness temperature for drought detection. Advances in Space Re-search 1995, 15, 91-100.

8- Title f table is missing?

Table 1. Statistical calculations NPP of slope change in different ecosystems

9- Rewrite the discussion part: Since there is limited (if any) access to ground data on NPP measurement in Iran...

There is limited (if any) access to ground data on NPP measurement in Iran. Therefore, it is difficult to conduct spatial matching and validate NPP estimates obtained through remote sensing. Previous researches (e.g., Running et al., 2000; Leeuw et al., 2019; Li et al., 2013; Wu et al., 2021) indicate that MODIS data have acceptable accuracy. MODIS sensor-based NPP data is available for only 46% of the areas of Iran’s ecosystems. There is no such data for Central Persian desert basins and South Iran Nubo-Sindian desert and semi-desert, which are characterized by a lack of vegetation. In these locations, the degree of carbon sequestration can be accessed via experimental models or ground observations. The average share of terrestrial carbon sequestration of plants in Iran was estimated to be 1.38 kg*C/m2/yr. This estimate is larger than the one proposed by Kamali et al. (2021), who studied a time period ending in 2014 and used average pixels.

10- Where is the correlation ? This is not the right way of presenting correlation??

According to the advice we got, instead of a map, use a map and analyze it

Reviewer 3 Report

I have read the manuscript entitled ‘Spatio-temporal analysis of carbon sequestration in different ecosystems of Iran and its relationship with agricultural droughts’. This is an excellent manuscript aimed to explore the spatiotemporal variations in NPP and its association with agricultural droughts in ecosystems over 20 years (2000-2020) in Iran. I would suggest minor revisions and give the chance to the authors to overlook the remaining errors in the manuscript.

Author Response

Hello and respect
The authors would like to thank the area editor and the reviewer for their precious time and invaluable comments. We have carefully addressed all the comments. The corresponding changes and refinements made in the revised paper are summarized in our response below.

1- Put comma before respectively. And do it throughout the manuscript

Southwestern Iran, respectively recorded the highest and lowest carbon sequestration rates.

Temperature, respectively which were extracted from different bands

And 225.99 g*C/m2, respectively

Correlations, respectively

2- VHI, please write the elaboration?

   Vegetation Health Index (VHI)

3- Add the implications of these results in one/two sentences?

Identification of areas with the greatest carbon sequestration changes could result in prioritizing varied ecosystems management for carbon sequestering. It can be also utilized in environmental planning such as scaling up ecosystems' values or estimating current and past ecological capacity.

3- Put comma here

Several million years, the fast carbon

4- Please define NPP here as net primary production (NPP)

NPP, which is defined as the net rate of carbon production in vegetation for a given period, indicates the mutual impact of environmental factors (e.g., soil and type of tree), human activities, and climatic factors (Yang et al., 2017; Hao et al., 2017).

5-who carried out the assessment?

Lieth and Whittaker (1975) Carried out the first NPP assessment on a global scale through conducting a regression analysis based on the data related

6- Comprised of?  The study area comprised Iran,     

The study area Iran, is located in Southeast Asia

7- Why not using only the abbreviation here? About annual Net Primary

About annual NPP at 500m pixel

8-   Is possible please use reference

It is the method of the authors of the article

10- Abbreviation at the bottom of the table

Standard Deviation

11- Single space ecosystem (Table1).

 (Table 1)

12- No significant OR Non significant

The whole article was replaced with the word non-significant

13- Please mention the title of the X and Y axes along with their units

According to the opinion of the other judges it was given as a map

14- Delete .             5. Conclusions

Reviewer 4 Report

This research paper (sustainability-2231830-peer-review-v1) is a case study of carbon sequestration in ecosystems to support drought affected agricultural systems of Iran. This paper is exclusive to the importance of environmental growth factors for the development of human related dynamic ecosystem that increases carbon cycle flux of plants. It is one of the suggestive contributions to understanding the agricultural droughts in ecosystems over last 20 years in Iran. Present research contributes to the climate change due to human activities that have influenced the dynamics of terrestrial ecosystem and regulates the primary source of fuel for energy in a global economy. This study emphasises the carbon cycle balance has under gone notable changes by the growing consumption of fossil fuels that release carbon into the atmosphere. This issue is most important to be addressed into the scientific community and must be published in Sustainability.

However, there are certain criticisms detected in this article that needs further amendment.

Some issues need to be clarified as follows:

1.    Present quantitative values of NPP in your study area.

2.    How NPP is related to the economic growth of Iran?

3.    What are the major causes of drought in Iran?

4.    In conclusions, discuss on the scientific methods that improve the situation of drought affected agriculture system in the Iran. 

I suggested this manuscript should be appropriate to recommend for publication in the Sustainability with minor revision.

Author Response

Hello and respect
The authors would like to thank the area editor and the reviewer for their precious time and invaluable comments. We have carefully addressed all the comments. The corresponding changes and refinements made in the revised paper are summarized in our response below.

  1. Present quantitative values of NPP in your study area.

Considering spatial extent, the climatic diversity distribution of NPP is very high in Iran particularly in North and Northwest parts of the country. In contrast, East and Southeast regions have lower NPP values such that Northern Forests NPP amounts to 1.3 and in the case of central deserts it is almost 0. During recent years, the major quantitively changes in NPP is caused by the pure effects of climate changes and intensified anthropogenic activities in Iran.

  1. How NPP is related to the economic growth of Iran?

Environmental degradation and land use alterations have been driven by Iran's economic growth which in turn influenced considerably ecological processes, including carbon sequestration. However, the relationship between NPP and the level of anthropogenic activities in Iran is very difficult to understand because most of Iran's production activities are concentrated in the central provinces like Tehran and Isfahan, while these provinces have high reliance of other regions on imports of raw materials and do not exploit local natural resources.

  1. What are the major causes of drought in Iran?

Drought is a serious environmental threat across Iran, mainly due to the climate change, however, its effects are exacerbated by rapid population growth, inappropriate population distributions, lands overharvesting, and poor water management (Madni, 2014).

  1. In conclusions, discuss on the scientific methods that improve the situation of drought affected in the Iran. 

Providing precise information for decision makers planning in order to make balance between society's needs and the exploitation of ecosystems' natural resources during drought periods could contribute to help PNN decreasing. Local non-governmental organizations (NGOs) and their participation in decision-making process, as well as information sharing via social media during drought periods play vital roles in this respect.

Round 2

Reviewer 1 Report

Accept in present form.

Author Response

Hello

 We sincerely thank the reviewer for taking the time to review our manuscript and providing constructive feedback to improve our manuscript.

1-   Delete "index". Add the full name of VHI.

2.2.2. Calculation of agricultural drought index

 Vegetation Health Index (VHI) combines...

2- From this map how a reader will understand what authors want to mean. There is no heading? No explanation

This map is the correlation percentage between NPP and agricultural drought. In Figure 6, this relationship was shown pixel by pixel, but here, to have a broader view and be able to give a general opinion, the average of that ecosystem area was obtained and compared with previous research. We have been explained in the discussion section.

Figure 7 shows the average of correlation between NPP and agricultural drought in different ecosystems. Various values indicate that the effect of droughts on NPP depends on the ecosystem type. These correlations could be associated the differences in drought intensity, drought duration, and vegetation type Pei et al. [76] and Yu et al. [77] discovered that the positive correlation between NPP and SPEI was more robust in the ecosystems located in arid and semi-arid climatic regions. Similarly, in the current study, the highest correlation was observed in the Zagros Mountains forest steppe (Figure 7), an ecosystem located in a semi-arid climate with the highest amount of rainfall in winter and spring. In contrast, the weakest correlation was spotted in the Registan-North Pakistan sandy desert with a cold desert climate. This ecosystem income-passes a small area in Iran; therefore, hence caution must be exercised in making claims based on the findings.

Reviewer 2 Report

Needs some more corrections

Author Response

(The authors gave the same response as above.)

Round 3

Reviewer 2 Report

It may be accepted
